# A Lesson for Sustainable Health Policy from the Past with Implications for the Future

**Göran Svensson [1], Rocio Rodriguez [1,2,*] and Carmen Padin [3]**

1   Marketing Department, Kristiania University College, 0107 Oslo, Norway; goran.svensson@kristiania.no
2   Marketing Department, University of Murcia, 30100 Murcia, Spain
3   Applied Economics Department, Universidade de Vigo, 36310 Vigo, Spain; padin@uvigo.es
*   Correspondence: rrodriguez@um.es

**Abstract:** Evidently, there are lessons to be learned on sustainable health policies from the SARS-CoV-2 pandemic. The past is a source of knowledge and experiences for the implementation and application of sustainable health policies in the future. This study has revealed doubts about the use of 7- and 14-days incidences, which have been applied as assessment approaches to the sustainable health policies used to control and monitor the SARS-CoV-2 pandemic across societies. Seven- and fourteen-day incidences have been used to determine measures and counter-measures against SARS-CoV-2 rather than infection rates. The research objective of this study was to assess the predictive abilities of infection rates versus 7- and 14-day incidences on SARS-CoV-2-related mortality and morbidity. The objective was also to assess the structural properties of a set of SARS-CoV-2-related variables. This study addressed the question of whether there is a lesson learned in terms of sustainable health policies on the use of 7- and 14-day incidences versus infection rates to predict SARS-CoV-2-related mortality and morbidity in a given context. We contend that there is at least one lesson to be learned on sustainable health policies from the SARS-CoV-2 pandemic. The infection rate was categorized as the independent manifest variable, as it is the one which is hypothesized to cause an effect on the outcome of the others in society regarding mortality and morbidity. Consequently, hospitalized patients, ICU patients and the deceased were categorized as dependent manifest variables. We tested the research model using Covariance-Based Structural Equation Modeling (CB-SEM) based on the first year of pandemic data before vaccines were used. This study indicates that the infection rates provided an enhanced predictability for SARS-CoV-2-related mortality and morbidity compared to 7- and 14-day incidences. The findings reported based on CB-SEM suggested that this has been a suitable way to assess the direct, indirect and mediating effects between a selection of SARS-CoV-2-related variables. We propose that our assessment approach to SARS-CoV-2 can be used as a complementary tool in decision-making on pandemic countermeasures to assess the health, social and economic costs of mortality and morbidity in a given context. We consider the finding that infection rates, rather than 7- and 14-day incidences, better predict SARS-CoV-2-related mortality and morbidity is a crucial lesson learned on sustainable health policies from the past, to be a crucial lesson for the future.

**Keywords:** sustainable; healthcare; health policy; pandemic; infection rate; 7-day incidence; 14-day incidence; intensive; care unit; hospitalization; deceased; patients





## 1. Introduction

Ioaniddes (2022; p. 1) [1] states, "There are no widely accepted, quantitative definitions for the end of a pandemic such as COVID-19." McCoy (2023; p. 1) [2] points out that "The end of a pandemic is as much a political act as biological reality". Furthermore, the awareness that a future pandemic may occur has become a concern. A revisit to assess the initial estimations of mortality and morbidity is therefore justified, as the time and timing of countermeasures are going to be crucial to monitor and control the emergence of future pandemics. The crucial question regarding the SARS-CoV-2 pandemic is therefore whether

the gathered knowledge and experiences have provided any lessons for sustainable health policies in the future.

There are two streams of studies reported in the literature. One stream focuses on using 7- and 14-day incidences to predict SARS-CoV-2 [3–5], while the other focuses on infection rates [6–8]. There are no studies that focus on comparing the predictive ability of 7- and 14-day incidences with that of infection rates on SARS-CoV-2-related mortality and morbidity in a given context.

The objective of this study focuses on the lessons that can be learned for sustainable health policies from the use of 7- and 14-day incidences versus infection rates to predict SARS-CoV-2-related mortality and morbidity in a given context. This study therefore assesses the predictive abilities of infection rates versus 7- and 14-day incidences for SARS-CoV-2-related mortality and morbidity. It also assesses the structural properties of a set of SARS-CoV-2-related variables. There is also no previous study that has assessed the direct, indirect and mediating effects between infection rates on the one side and mortality and morbidity on the other. The structural properties have not been prioritized but taken for granted in terms of the idea that infection rates influence mortality and morbidity, nor have their direct, indirect and mediating effects.

The pandemic revealed the importance of controlling and closely monitoring SARS-CoV-2 across societies. There has been widespread and ongoing use in sustainable health policies of 7- and 14-day incidences to determine countermeasures against SARS-CoV-2. We argue that it is a questionable use that neglects the empirical evidence reported in this study, namely that the infection rate (i.e., the ratio between the number of tests and positive cases) correlates much stronger with SARS-CoV-2-related mortality and morbidity, ranging between 0.874 and 0.937, while 7- and 14-day incidence rates range between 0.456 and 0.666. We contend that this finding is a relevant and important lesson for establishing sustainable health policies from the past and will be a valuable lesson for the future. It implies that infection rates provide an enhanced validity and reliability in predicting SARS-CoV-2-related mortality and morbidity in comparison to 7- and 14-day incidences.

Consequently, this study contributes to demonstrating that the infection rate of SARS-CoV-2, rather than 7- and 14-day incidences, enhances the predictive abilities for SARS-CoV-2-related mortality and morbidity. This study contributes also to making predictions based on Covariance-Based Structural Equation Modeling (CB-SEM) on the numbers of intensive care unit (ICU) and hospitalized patients, as well as of the deceased, according to infection rates.

The structural properties (i.e., causes and effects) between the infection rate and ICU and hospitalized patients as well as the deceased have still not been explicitly explored based on the past as a lesson for sustainable health policies in the future. In particular, the direct, indirect and mediating effects have still not been tested empirically and thus not modeled statistically. One of the research objectives of this study was therefore to assess the structural properties of a set of SARS-CoV-2-related variables. The aim was to shed light on the extent to which and how the infection rates of SARS-CoV-2 could model the structural properties of cause and effect on mortality and morbidity in given society.

The research model consisted of four SARS-CoV-2-related variables (see Figure 1) as follows: (i) infection rate; (ii) hospitalized patients; (iii) ICU patients; (iv) the deceased. The research model enabled an assessment of the structural properties (i.e., direct, indirect and mediating effects) between them.

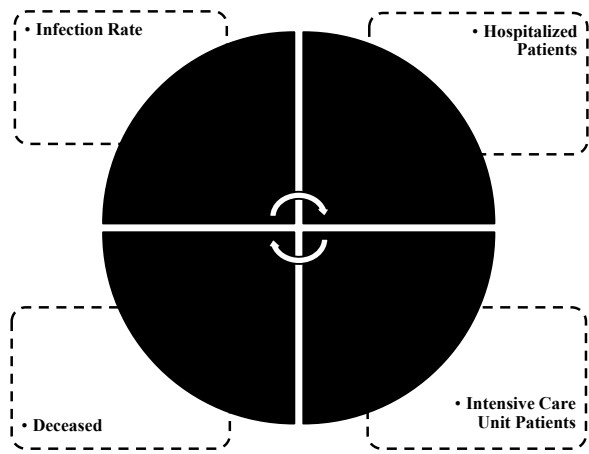

**Figure 1.** Research model.

## 2. Research Context

The pandemic caused by SARS-CoV-2 revealed the importance of controlling and closely monitoring the infection rate across societies. The empirical evidence reported in this study shows that the infection rate correlates much stronger with mortality and morbidity compared to 7- and 14-day incidence rates. It implies that the infection rate provides an enhanced validity and reliability in predicting SARS-CoV-2 mortality and morbidity. Three criteria were applied to undertake this study in Sweden, namely (i) the transparency of the databases; (ii) the availability of valid and reliable data; and (iii) officially verified data being in the databases. Three separate databases were subsequently identified containing relevant COVID-19 data for this study. Although this study is limited to Sweden, its results are most likely universal, as the effect of SARS-CoV-2 infection rates on mortality and morbidity across countries was drastic before vaccines were introduced. The effect on mortality and morbidity was not the same across countries, though the countermeasures undertaken were different. However, national borders are not likely to alter the core structural properties of the research model tested in this study.

## 3. Methods

The pandemic and epidemic research context of this study was Sweden. The Swedish approach to handling SARS-CoV-2 stood out particularly in the first wave [9] and has been conspicuous internationally [10,11]. It was therefore a relevant point of reference and a benchmark to assess in relation to other countries.

The implementation of the Swedish strategy contained fewer countermeasures applied to the public compared to many other countries [12], such as Norway, Finland and Denmark, but also in comparison to the majority of countries in Europe. The mortality rate in Sweden per capita was among the top in the world [13,14] despite the fact that it is a developed economy with a small population and a low density of population [15].

An important observation is that a large part of the Swedish population is culturally and socially characterized by social distancing. The social and cultural context of Norway, Finland and Denmark resembles Sweden to a major extent [16–18]. Economic welfare is also high across these countries [15]. However, the governments and agencies of public health in Norway, Finland and Denmark acted in a different way compared to in Sweden, with stricter and more intrusive countermeasures to manage the infection rate of SARS-CoV-2 in the society to protect their populations.

An assessment of the healthcare statistics between Sweden and Norway, Finland and Denmark related to SARS-CoV-2 demonstrated substantial differences [13,14]: (i) the rate of verified cases was tripled per capita in Sweden than in Norway, Finland and Denmark altogether; (ii) the rate of undertaken tests per capita in Norway, Finland and Denmark together was septuple compared to Sweden; (iii) the rate of mortality was sextupled per

capita in Sweden compared to Norway, Finland and Denmark altogether. Consequently, the morbidity per capita (i.e., the number of ICU and hospitalized patients) was higher in Sweden as well than in its neighboring countries.

The COVID-19 data in this study originate from the Agency of Public Health [19], the National Board of Health and Welfare and the Swedish Intensive Care Registry—SIR [20,21]. The databases were regularly followed up and updated during the first year of the pandemic. We tested the research model outlined in Figure 1 using Covariance-Based Structural Equation Modeling (CB-SEM) and using the SPSS Amos 26.0 software. CB-SEM enabled us to shed light on the direct, indirect and mediating effects between the SARS-CoV-2-related variables.

CB-SEM is a multivariate technique that enables us to verify the validity and reliability of the structural (and measurement) properties of a research model. It is a so-called full-information technique that considers the co-variances simultaneously between the variables tested in the research model. CB-SEM enables us to test whether there are direct and indirect effects between variables, as well as the existence of mediating effects between the same variables.

The research model was therefore divided into three sub-models: (i) direct effects; (ii) direct and indirect effects; (iii) mediating effects. The infection rate was categorized as the independent manifest variable, as it is the one which is hypothesized to cause an effect on the outcome of the others in society. Consequently, ICU patients, hospitalized patients and the deceased were categorized as dependent manifest variables.

Naturally, the rate of infection at a specific moment does not commonly cause hospitalization and ICU stays as well deaths in the moment (i.e., there is a delay in time). A delay in time between COVID-19-related variables is subsequently used in the CB-SEM analyses. An example could be a confirmed case today (day 1), them being hospitalized tomorrow (day 2), them being entered into the ICU the day after tomorrow (day 3) and them dying the following day (day 4). However, these could happen on the same day, but, most commonly, there is a delay in time between these events.

In fact, a delay in time commonly taking place after a SARS-CoV-2 case was assessed and confirmed, as was the following impact on the number of ICU and hospitalized patients. A delay in time also occurred between the infection and death of patients. In fact, the delay in time in Sweden between the moment of confirmed infection and (i) the hospitalization of patients was a mean of 6.2 days [20]; (ii) patients being entered into the ICU was a mean of 10.6 days [20]; and (iii) patients dying was a mean value of 12.5 days [20]. The SARS-CoV-2-related variables in the research model were therefore analyzed with a time delay of one week between the infection rate and the other SARS-CoV-2-related variables.

## 4. Results

The empirical evidence in this study showed that the infection rate (i.e., the ratio between the number of tests and positive cases) correlates much stronger with SARS-CoV-2 mortality and morbidity, ranging between 0.874 and 0.937, while 7- and 14-day incidence rates range between 0.456 and 0.666. This section therefore reports the empirical findings based on the infection rate based on Swedish SARS-CoV-2-related data for 294 days (i.e., 42 weeks) in the period from early March (2020) to late December (2020) [11]. The timespan of the data analysis begins on March 9 (week 11), when the first COVID-19 death was registered [11], and finishes on December 27 (week 52), when vaccinations started [22], to avoid biases in the data analyses, with the following variables: (i) average infection rate per week; (ii) total number of hospitalized patients per day; (iii) total number of ICU patients per day; (iv) total number of deceased per day.

There were highly significant correlations between the SARS-CoV-2-related variables, with the correlation coefficients ranging from 0.835 to 0.914 with *p*-values at 0.000. We therefore tested six hypothesized relationships between the SARS-CoV-2-related variables in the research model displayed in Figure 1:

**Hypothesis 1 (H$_1$):** *Infection rate related positively to hospitalized patients.*

**Hypothesis 2 (H$_2$):** *Infection rate related positively to ICU patients.*

**Hypothesis 3 (H$_3$):** *Infection rate related positively to the deceased.*

**Hypothesis 4 (H$_4$):** *Hospitalized patients related positively to the deceased.*

**Hypothesis 5 (H$_5$):** *ICU patients related positively to the deceased.*

**Hypothesis 6 (H$_6$):** *Hospitalized patients related positively to ICU patients.*

The regression statistics and significance levels of the hypothesized relationships are reported in Table 1 and divided into direct effects, direct and indirect effects and mediating effects. Table 1 also reports the CB-SEM goodness-of-fit measures.

**Table 1.** Regression and goodness-of-fit measures.

| Regression Statistics | | | | | |
|---|---|---|---|---|---|
| **(i) Direct Effects** | | | | | |
| **Hypothesis** | **Exogenous Variable** | **Endogenous Variable** | **Regression Coefficients** | **Significance** | **Results** |
| 1 | Infection Rate | Hospitalized Patients | 0.914 | 0.000 | Supported |
| 2 | Infection Rate | ICU Patients | 0.891 | 0.000 | Supported |
| 3 | Infection Rate | Deceased | 0.896 | 0.000 | Supported |
| **(ii) Direct and Indirect Effects 1** | | | | | |
| 1 | Infection Rate | Hospitalized Patients | 0.914 | 0.000 | Supported |
| 2 | Infection Rate | ICU Patients | 0.891 | 0.000 | Supported |
| 3 | Infection Rate | Deceased | −0.042 | 0.433 | Not Supported |
| 4 | Hospitalized Patients | Deceased | 0.755 | 0.000 | Supported |
| 5 | ICU Patients | Deceased | 0.281 | 0.000 | Supported |
| **(iii) Direct and Indirect Effects 2** | | | | | |
| 1 | Infection Rate | Hospitalized Patients | 0.914 | 0.000 | Supported |
| 2 | Infection Rate | ICU Patients | 0.831 | 0.000 | Supported |
| 4 | Hospitalized Patients | Deceased | 0.728 | 0.000 | Supported |
| 5 | ICU Patients | Deceased | 0.264 | 0.000 | Supported |
| 6 | Hospitalized Patients | ICU Patients | 0.066 | 0.326 | Not Supported |

**Table 1.** *Cont.*

| Regression Statistics | | | | | |
|---|---|---|---|---|---|
| **(i) Direct Effects** | | | | | |
| **Hypothesis** | **Exogenous Variable** | **Endogenous Variable** | **Regression Coefficients** | **Significance** | **Results** |
| **(iv) Direct and Indirect Effects 3** | | | | | |
| 1 | Infection Rate | Hospitalized Patients | 0.914 | 0.000 | Supported |
| 2 | Infection Rate | ICU Patients | 0.833 | 0.000 | Supported |
| 3 | Infection Rate | Deceased | −0.041 | 0.436 | Not Supported |
| 4 | Hospitalized Patients | Deceased | 0.751 | 0.000 | Supported |
| 5 | ICU Patients | Deceased | 0.281 | 0.000 | Supported |
| 6 | Hospitalized Patients | ICU Patients | 0.064 | 0.336 | Not Supported |
| **(v) Mediating Effects** | | | | | |
| 1 | Infection Rate | Hospitalized Patients | 0.914 | 0.000 | Supported |
| 2 | Infection Rate | ICU Patients | 0.891 | 0.000 | Supported |
| 4 | Hospitalized Patients | Deceased | 0.731 | 0.000 | Supported |
| 5 | ICU Patients | Deceased | 0.263 | 0.000 | Supported |

| Goodness-of-Fit Statistics | | | | | | | | | |
|---|---|---|---|---|---|---|---|---|---|
| **Model** | **Chi-square** | **Degrees of Freedom** | **Significance** | **NFI** | **CFI** | **RMSEA** | **PRATIO** | **PNFI** | **PCFI** |
| Direct Effect | 252.151 | 3 | 0.000 | 0.853 | 0.854 | 0.532 | 0.300 | 0.256 | 0.256 |
| Direct/Indirect Effects 1 | 0.851 | 1 | 0.356 | 1.000 | 1.000 | 0.000 | 0.100 | 0.100 | 0.100 |
| Direct/Indirect Effects 2 | 0.611 | 1 | 0.434 | 1.000 | 1.000 | 0.000 | 0.100 | 0.100 | 0.100 |
| Direct/Indirect Effects 3 | 0.000 | 0 | NA * | 1.000 | 1.000 | 0.762 | 0.000 | 0.000 | 0.000 |
| Mediating Effects | 1.498 | 2 | 0.473 | 0.999 | 1.000 | 0.000 | 0.200 | 0.200 | 0.200 |

\* Not applicable.

### 4.1. Direct Effects

We tested the direct effect of three hypothesized relationships in the research model, based on the infection rate in relation to ICU patients, hospitalized patients and the deceased (i.e., Hypotheses 1–3), as displayed in Figure 2.

The hypothesized relationships in the research model were highly satisfactory but limited to direct effects, as illustrated in Figure 2. They were all significant at $p = 0.000$ with the significant standardized regression weights ranging between 0.891 and 0.914, as illustrated in Table 1 (i.e., direct effects).

However, the goodness-of-fit measures of the structural model, as shown in Table 1, were unsatisfactory in testing the direct effects. The Chi-square is 252.151 with three degrees of freedom and is statistically significant at $p = 0.000$, with a normed Chi-square ($X^2/\mathrm{df}$) of 84.050. The fit statistics were as follows: NFI is 0.853 and CFI is 0.854. The RMSEA is 0.532 with a 90% confidence interval of 0.477–0.589. The unsatisfactory fit of the model indicated the existence of indirect effects between the SARS-CoV-2-related variables.

Consequently, the hypothesized relationships (direct effects) of the infection rate with ICU patients, hospitalized patients and the deceased, as displayed in Figure 2, were all highly significant as expected, but the goodness-of-fit measures were unsatisfactory. We therefore tested three options for the direct and indirect effects in the subsequent models, displayed in Figure 3.

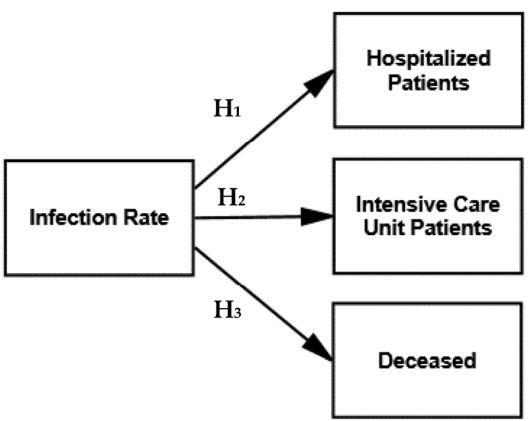

**Figure 2.** Research model—D = direct effects.

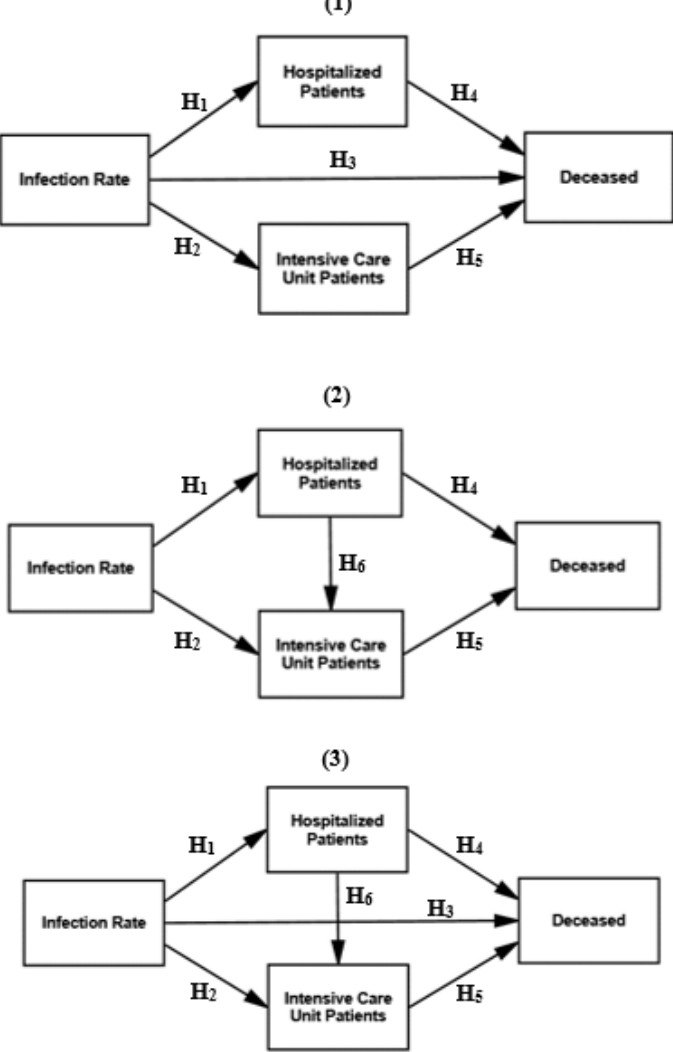

**Figure 3.** Research models—direct and indirect effects 1–3.

### 4.2. Direct and Indirect Effects

We tested up to six hypothesized relationships of the direct and indirect effects in models 1–3, as displayed in Figure 3.

The hypothesized relationships, as illustrated in Figure 3, were significant for four out of six at $p = 0.000$, except between infection rate and the deceased as well as between hospitalized patients and ICU patients (regression weights ranging between $-0.041$ and 0.066 at $p = 0.326$–0.436), with the significant standardized regression weights ranging between 0.263 and 0.914, as reported in Table 1 (i.e., direct and indirect effects 1–3).

The hypothesis $H_3$ (i.e., infection rate relates to the deceased) was not significant in models 1 and 3 (see Figure 3). The hypothesis $H_6$ (i.e., ICU patients relates to hospitalized patients) was also not significant in models 2 and 3 (see Figure 3). The goodness-of-fit measures (see the lower part of Table 1) for models 1–3 were mostly satisfactory but have revealed that two hypothesized relationships should be excluded (i.e., $H_3$ and $H_6$).

Consequently, models 1–3 testing the direct and indirect effects demonstrate that there was, to some extent, an unexpected and also unrevealed mediating effect, namely that hospitalized patients and ICU patients mediate the effect between the infection rate and the deceased. The mediating effect was therefore tested in the next section.

### 4.3. Mediating Effects

We tested the research model on the direct effects of the infection rate on hospitalized patients and ICU patients but excluding a direct effect on the deceased. We also tested the mediating effect of hospitalized and ICU patients on the relationship between the infection rate and the deceased, as shown in Figure 4.

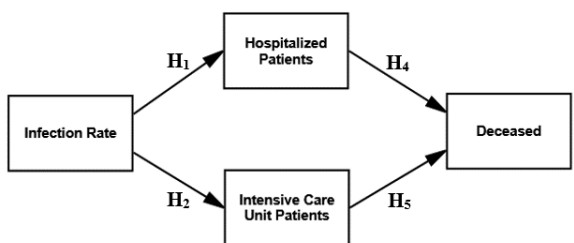

**Figure 4.** Research model—mediating Effect.

The goodness-of-fit measures of the structural model testing the direct, indirect and mediating effects were satisfactory, with a Chi-square of 1.498 with two degrees of freedom. The Chi-square was not significant at $p = 0.473$, with a normed Chi-square ($X^2/df$) at 0.749. Nevertheless, the NFI is 0.999, and the CFI is 1.000. The RMSEA is 0.000 with a 90% confidence interval ranging between 0.000 and 0.106. All the CB-SEM measures were in line with the recommended thresholds.

Furthermore, the parsimony-adjusted measures were as follows: PRATIO at 0.200, PNFI at 0.200 and PCFI at 0.200, being higher compared to models of the direct and indirect effects 1–3 (see the lower part of Table 1). It suggested the comparatively better fit of the research model considering the mediating effect.

The mediating effect also reduced the regression coefficient of the hypothesized relationship between the infection rate and the deceased from 0.896 (significant at 0.000) to $-0.041$ (significant at 0.436). The regression coefficient between hospitalized patients and ICU patients also decreased from 0.835 (significant at 0.000) to 0.066 (significant at 0.326). The hypothesized relationships of the research model, as illustrated in Figure 4, were all significant at $p = 0.000$, with the significant standardized regression weights ranging between 0.263 and 0.914, as illustrated in Table 1 (i.e., mediating effects).

Indeed, the revealed mediating effect unveiled was logically expected to some extent, but that the direct effect between the infection rate and the deceased disappears completely was unexpected. A tentative explanation was that not all the deceased were

hospitalized or admitted to ICUs but died in old age homes or at home without advanced medical treatment.

## 5. Lesson for the Future, Implications and Further Research

The results reported based on CB-SEM suggest a lesson for the future, namely that the approach based on infection rate, rather than 7- and 14-day incidence rates, shown in Figure 5 is suitable for sustainable health policies in assessing the direct, indirect and mediating effects between a selection of SARS-CoV-2-related variables (i.e., between infection rate and hospitalized and ICU patients as well as the deceased). Furthermore, the lesson indicates that the approach undertaken in this study enables the possibility of foresight of SARS-CoV-2-related mortality and morbidity in advance based on the current infection rates.

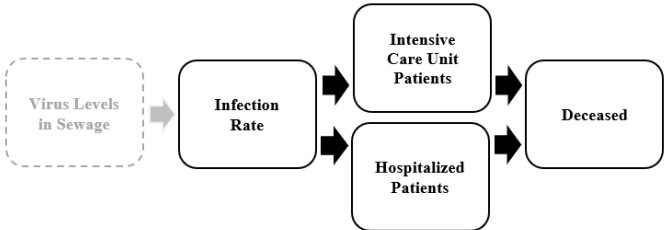

**Figure 5.** An assessment approach for SARS-CoV-2.

An important implication is that the CB-SEM reveals the structural properties between the studied SARS-CoV-2-related variables. The structural properties visualize a pattern of SARS-CoV-2-related consequences in society that can be remedied using the appropriate preventive measures and restrictions in sustainable health policies. One opportunity for further research is to expand the approach to assessing the direct and indirect effects of the studied SARS-CoV-2-related variables on other healthcare-related variables and resources, such as medical treatments, equipment and professionals.

The CB-SEM models in Figures 2–4 and the related statistics based on SARS-CoV-2-related variables indicate highly significant relationships between the infection rate on the one side and hospitalized patients, ICU patients and the deceased on the other. However, the CB-SEM models of the direct, indirect and mediating effects reveal that hospitalized patients and ICU patients mediate the effect between the infection rate and the deceased. The CB-SEM shows that although there is a highly significant direct effect between the infection rate and the deceased, the effect becomes non-significant and non-existent when hospitalized patients and ICU patients are introduced as mediators.

The CB-SEM model displayed in Figure 4 demonstrates, as expected, that the severity of morbidity, not the infection rate per se, influences the mortality in the end. It thus demonstrates that the infection rate is the crucial variable to monitor and control before the escalation of mortality and morbidity. If monitoring and control fails, the infection rate increases exponentially, exposing the downside of reactive measures and restrictions. The infection rate is therefore the inevitable SARS-CoV-2-related variable that must be taken into account in any assessment approach. The infection rate is therefore also an inevitable cause to benchmark in sustainable health policies regarding preventive measures and restrictions instead of 7- and 14-day incidence rates in the future. It is thus advisable for sustainable health policies of pandemic plans to have pre-established and stipulated protocols of when, where and how to act and what to reach.

The assessment approach derived from the CB-SEM models clearly demonstrates that the number of patients in need of hospitalization and ICU admission is related to mortality in a particular society. It should be noted that improved medical treatments and the availability of medicines and vaccines will weaken the effect of the mediators on the relationship between the infection rate and the deceased, but the structural properties

are likely to remain intact. They are also likely to be universal and applicable in other national settings.

An assessment approach based on the infection rate, rather than 7- and 14-day incidence rates, described for SARS-CoV-2 can be used in sustainable health policies as a future complementary tool in decision-making to assess the health, social and economic costs of mortality and morbidity in a given context. This assessment approach facilitates the continuous use of emerging knowledge about the direct, indirect and mediating effects as early as possible. It requires that the implementation of pandemic plans undergoes continuous revision and the updating of protocols in sustainable health policies so as to impose preventive measures and restrictions of when, where and how to act and what to reach on time. The assessment approach both encourages and stresses the proactive use of epidemic data sources and resources.

A potential extension of the assessment approach is available at the regional or municipal level, linking analyses that monitor the SARS-CoV-2 levels in sewage (i.e., waste water) at treatment plants to estimate community infection rates (i.e., not the time-delayed 7- and 14-day incidence rates), as shown in Figure 5. This was beyond the scope and reach of this study, as these data are not available at the national level, only for a few larger cities. Nevertheless, we hypothesize that there is most likely a significant relationship between the virus levels in sewage and the infection rate in an extended assessment approach [23,24].

Nevertheless, there is an opportunity for further research to test the direct, indirect and mediating effects of the SARS-CoV-2 levels in sewage on the one side and hospitalized and ICU patients as well as the deceased on the other hand, with the infection rate as mediator. Tentatively, the SARS-CoV-2 levels in sewage enable us to estimate the outcomes on the SARS-CoV-2-related variables tested in this study. Such waste water analyses may serve as early warnings of SARS-CoV-2 before infection rates appear in the healthcare sector. We contend that it is therefore an additional SARS-CoV-2-related variable that may enhance the assessment of sustainable health policies to proactively monitor and control the outcomes of mortality and morbidity based on the infection rate.

## 6. Limitations

This study has several limitations, such as that the COVID data gathered are limited to just one country (i.e., Sweden). The time frame of the data is also limited to the period before vaccines were introduced, although this was chosen to keep the data unbiased. Another limitation is the absence of other COVID-related data that were not available at the time, such as the virus levels in sewage. In addition, an assessment of sociographic and psychographic variables was not possible given that data were not available that could have yielded additional insights from the population characteristics. This study is also limited to one pandemic, so comparisons are not possible. Limitations also relate to the type of the virus and the effect that this had on the morbidity and mortality of populations. Although this study does have these limitations, it nonetheless offers a lesson from the past for the future and also provides suggestions for further research.

## 7. Concluding Thoughts

We conclude that the assessment approach of this study possesses several generic leverage effects of sustainable health policies, such as:

(i) A foundation on which to establish a pandemic plan with stipulated preventive measures and restrictions to provide guidance as to where, when and how to handle outbreaks of virulent viruses (such as MERS, SARS-CoV-1 and SARS-CoV-2);

(ii) Examines the resource requirements, proactively considering the outcomes of the numbers of hospitalized and ICU patients, as well as the deceased, at different levels of infection rates;

(iii) Provides a basis for establishing break-even points between the health, social and economic consequences of preventive measures and restrictions;

(iv)   Assesses the efficiency and effectiveness of the measures and restrictions imposed by governments and public health agencies;

(v)    Reveals the ethical and moral boundaries regarding acceptable mortality and morbidity;

(vi)   Complements the methodological use of 7- and 14-day incidence rates, epidemic scenarios and epidemic curves.

A reliance on epidemic scenarios can disregard what is actually happening in society. A scenario is a prediction based on a set of assumptions. The prediction of an epidemic scenario that is not based on empirically verified structural properties outlining the direct, indirect and mediating effects between SARS-CoV-2-related variables becomes questionable in terms of both its validity and reliability. It was therefore not uncommon for epidemic scenarios during the beginning of the pandemic to be inaccurate in predicting the forthcoming infection rates of SARS-CoV-2 and the numbers of hospitalized and ICU patients as well as of the deceased. It was therefore also common that preventive measures and restrictions in sustainable health policies became reactive instead of proactive.

Similarly, reliance on the epidemic curve and 7- and 14-day incidence rates as points of reference in sustainable health policies were per se also reactive rather than proactive. Precious time was potentially lost, with preventive measures and restrictions being implemented too late. Knowledge about the direct, indirect and mediating effects between SARS-CoV-2-related variables provides guidance for establishing and imposing efficient (i.e., least use and waste of resources) and effective (i.e., degree of achieving the desired results) preventive measures and restrictions.

The lessons based on the assessment approach described for SARS-CoV-2 can be used in sustainable health policies as a complementary tool in decision-making to assess the health, social and economic costs of mortality and morbidity in a given context. The assessment approach required continuously using emerging knowledge about the direct, indirect and mediating effects as early as possible (i.e., daily instead of weekly). It requires that the implementation of pandemic plans undergoes continuous revision and the updating of protocols in sustainable health policies so as to impose preventive measures and restrictions of when, where and how to act and what to reach on time. This assessment approach both encourages and stresses the proactive use in sustainable health policies of epidemic data sources and resources.

We have addressed the question regarding the SARS-CoV-2 pandemic of whether the gathered knowledge and experiences from the use of 7- and 14-day incidences of SARS-CoV-2 rather than infection rates to predict mortality and morbidity in a given context have provided a lesson for sustainable health policies in the future. We contend that there is at least one lesson to be learned, consisting of several constituents of sustainable health policies, from the emergence and outbreak of the SARS-CoV-2 pandemic in early 2020. This study focused on the lessons that can be learned from the use of the infection rates of SARS-CoV-2 rather than the 7- and 14-day incidences of SARS-CoV-2 to predict mortality and morbidity in a given context. We conclude that that the infection rate enhances the predictability of mortality and morbidity. In fact, we consider this to be a crucial lesson learned in sustainable health policies from the past as a lesson for the future.

**Author Contributions:** Conceptualization, G.S. and R.R.; validation, R.R.; investigation, C.P.; writing—original draft, G.S. and R.R. All authors have read and agreed to the published version of the manuscript.

**Funding:** This research received no external funding.

**Institutional Review Board Statement:** Ethical approval was not required for this study, as we worked with anonymized and dissociated public data.

**Informed Consent Statement:** Not applicable.

**Data Availability Statement:** No new data were created in this study.

**Conflicts of Interest:** No potential competing interests were reported by the authors.

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
