# Peer review of "A Lesson for Sustainable Health Policy from the Past with Implications for the Future"

_sustainability, doi:10.3390/su16051778_

Round 1
Reviewer 1 Report
Comments and Suggestions for Authors
The article evaluates the effectiveness of 7- and 14-day incidence rates versus infection rates in predicting SARS-CoV-2 related morbidity and mortality. The authors use Covariance-Based Structural Equation Modeling (CB-SEM) to analyze data from the first year of the pandemic in Sweden, before vaccines were implemented. My suggestions before publishing include the following:
- Please clarify the criteria for selecting the Swedish data set and whether its findings can be generalized to other populations or settings.
- Please consider elucidating more on the CB-SEM approach, especially for readers unfamiliar with this method. I think explaining how this model contributes explicitly to the accuracy of the predictions would be beneficial.
- I suggest expanding on how the findings of this study compare with other international studies. This would provide a global context and help understand the Swedish experience's uniqueness.
- Please include a detailed discussion on the study's limitations and potential areas for future research. This would help readers comprehend the scope and potential impact of the study.
- I would recommend adding a section detailing the specific policy implications of these findings to make sure it's clear. This would make the practical applications of the study more explicit for policymakers and public health officials.
No comment
Author Response
REVISION REPORT: JBIM-05-2023-0245.R1
Title: “Lessons for Sustainable Health Policy and the Past as Lessons for the Future”
Dear Editor, Associate Editor and Reviewers,
We are pleased to have the opportunity to make a revision of our paper for further consideration in Sustainability. We provide a detailed response (IN YELLOW) to each reviewer’s comments (IN GREEN).
We hope that you will find our revised paper much improved and satisfactory. Please note that we highlight major changes and amendments in the revised paper IN BLUE.
We believe that we have been able to address and justify our responses to all review comments appropriately.
Best regards,
The authors
Reviewer 1:
The article evaluates the effectiveness of 7- and 14-day incidence rates versus infection rates in predicting SARS-CoV-2 related morbidity and mortality. The authors use Covariance-Based Structural Equation Modeling (CB-SEM) to analyze data from the first year of the pandemic in Sweden, before vaccines were implemented.
The comments have been noted, thanks.
My suggestions before publishing include the following:
- Please clarify the criteria for selecting the Swedish data set and whether its findings can be generalized to other populations or settings.
We write as follows:
Three criteria were applied to undertake this study in Sweden, namely: (i) transparency of data bases; (ii) availability of valid and reliable data; and (iii) officially verified data in data bases. Three separate data bases were subsequently identified containing relevant COVID-19 data for this study. Although this study is limited to Sweden its results are most likely universal as the effect of SARS-CoV-2 infection rates on morbidity and mortality across countries was drastic before vaccines were introduced. The effect on morbidity and mortality was not the same across countries, though to the countermeasures undertaken were different. However, national borders were not likely to alter the core structural properties of the research model tested in this study.
- Please consider elucidating more on the CB-SEM approach, especially for readers unfamiliar with this method. I think explaining how this model contributes explicitly to the accuracy of the predictions would be beneficial.
We write as follows:
CB-SEM is a multivariate technique that enables to verify the validity and reliability of measurement and structural properties of a research model. It is a so-called full-information technique that considers the co-variances simultaneously between variables tested in the research model. CB-SEM enables to test whether there are direct and indirect effect between variables, as well as the existence of mediating effects between the same variables.
- I suggest expanding on how the findings of this study compare with other international studies. This would provide a global context and help understand the Swedish experience's uniqueness.
Ioaniddes (2022; p1) states “There are no widely accepted, quantitative definitions for the end of a pandemic such as COVID- 19. McCoy (2023; p.1) points out that: “The end of a pandemic is as much a political act as biological reality”. Furthermore, the awareness of a future pandemic may occur has become a concern. A revisit to assess the initial estimations of morbidity and mortality is therefore justified as time and timing of countermeasures are going to be crucial to monitor and control the emergence of future pandemics. The crucial question regarding the SARS-CoV-2 pandemic is therefore whether gathered knowledge and experiences have provided any lesson for sustainable health policies in the future.
There are two streams of studies reported in literature. One stream focuses on 7- and 14-days incidences to predict SARS-CoV-2 (e.g., Byambasuren et al., 2020; Sanjuán,2021; Scobie et al., 2021), while the other focuses on infection rates (e.g., Behnood et al., 2020; Manski and Molinari, 2021; Salgotra et al., 2020). To the best of the authors knowledge there are no studies that focus on comparing the predictive ability between 7- and 14-days incidences and infection rates on SARS-CoV-2 related morbidity and mortality in a given population.
The objective of this is study focuses on the lesson that can be learnt for sustainable health policies from the use of 7- and 14-days incidences versus infection rates to predict SARS-CoV-2 related morbidity and mortality in a given population. This study therefore assesses the predictive abilities of infection rates versus 7- and 14-days incidences on SARS-CoV-2 related morbidity and mortality. It also assesses the structural properties of a set of SARS-CoV-2 related variables. To the authors knowledge, there is also no previous study that has assessed the direct, indirect and mediating effects between infection rates on the one side, and morbidity and mortality on the other. The structural properties have not been prioritized, but taken for granted in terms of that infection rates influence morbidity and mortality, not the direct, indirect and mediating effects.
- Please include a detailed discussion on the study's limitations and potential areas for future research. This would help readers comprehend the scope and potential impact of the study.
Potential areas for future research are presented in the end section. We have made the following amendment regarding study limitations:
This study suffers from several limitations, such as that the COVID data gathered is limited to one country (i.e., Sweden), time frame of data (i.e., before vaccines were introduced), and other COVID data (e.g., virus levels in sewage) were not available. Though this study contains limitations, it offers a lesson from the past for the future and also suggestions for further research.
- I would recommend adding a section detailing the specific policy implications of these findings to make sure it's clear. This would make the practical applications of the study more explicit for policymakers and public health officials.
We write as follows in the end section:
We conclude that the assessment approach of this study possesses several generic leverage effects of sustainable health policies, such as:
(i) a foundation to establish a pandemic plan with stipulated preventive measures and restrictions to provide guidance as to what, where, when and how to handle the outbreaks of virulent viruses (such as SARS-CoV-2, SARS-CoV-1 and MERS);
(ii) examines the resource requirement, proactively considering the outcome of hospitalized and intensive care unit patients, as well as deceased at different levels of infection rates;
(iii) provides a basis for establishing break-even points between health, social and economic consequences of preventive measures and restrictions;
(iv) assesses the efficiency and effectiveness of imposed measures and restrictions by governments and public health agencies;
(v) reveals the ethical and moral boundaries regarding acceptable morbidity and mortality; and finally
(vi) complements the methodological use of 7- and 14-days incidence rates, epidemic scenarios and epidemic curves.

Reviewer 2 Report
Comments and Suggestions for Authors
Dear Authors,
Based on your manuscript titled "Lessons for Sustainable Health Policy and the Past as Lessons for the Future," I have a few comments on changes that should be made to this article.
1. Methods - you did not mention any specific tests that were used or any statistical metrics. You did not describe the database used in the study. Please provide more details. The entire section in the methods subsection is irrelevant. What hypotheses were set in this study and why?
2. Title and purpose of the study - in my opinion, you did not emphasize the purpose in the introduction and discussion.
3. Abstract and introduction - there is often repetition in the abstract and introduction section. This should be corrected, and the introduction should provide more information about the background of the study and the literature review in this area. Definitions of terms used, such as " generic research model" should be added.
4. Figures - bad quality.
5. From a researcher's perspective, I have to ask this: Why do you want to publish an article in 2024 based on results from 2021? I did not notice "a lesson for the future" in your manuscript (discussion).
Comments on the Quality of English LanguageEnglish needs to be improved
Author Response
REVISION REPORT: JBIM-05-2023-0245.R1
Title: “Lessons for Sustainable Health Policy and the Past as Lessons for the Future”
Dear Editor, Associate Editor and Reviewers,
We are pleased to have the opportunity to make a revision of our paper for further consideration in Sustainability. We provide a detailed response (IN YELLOW) to each reviewer’s comments (IN GREEN).
We hope that you will find our revised paper much improved and satisfactory. Please note that we highlight major changes and amendments in the revised paper IN BLUE.
We believe that we have been able to address and justify our responses to all review comments appropriately.
Best regards,
The authors
Reviewer 2:
Dear Authors,
Based on your manuscript titled "Lessons for Sustainable Health Policy and the Past as Lessons for the Future," I have a few comments on changes that should be made to this article.
- Methods - you did not mention any specific tests that were used or any statistical metrics.
SEM is a multivariate statistical technique of analysis that contains several tests that are performed, such as correlation tests, factor analysis and regression analysis. It is commonly not made explicit in research reported based on CB-SEM. Nevertheless, multiple statistical estimates are reported as done in common practice of CB-SEM.
We have made an amendment as follows:
CB-SEM is a multivariate technique that enables to verify the validity and reliability of measurement and structural properties of a research model. It is a so-called full-information technique that considers the co-variances simultaneously between variables tested in the research model. CB-SEM enables to test whether there are direct and indirect effect between variables, as well as the existence of mediating effects between the same variables.
- You did not describe the database used in the study.
We write as follows:
Three criteria were applied to undertake this study in Sweden, namely: (i) transparency of data bases; (ii) availability of valid and reliable data; and (iii) officially verified data in data bases. Three separate data bases were subsequently identified containing relevant COVID-19 data for this study. Although this study is limited to Sweden its results are most likely universal as the effect of SARS-CoV-2 infection rates on morbidity and mortality across countries was drastic before vaccines were introduced. The effect on morbidity and mortality was not the same across countries, though to the countermeasures undertaken were different. However, national borders were not likely to alter the core structural properties of the research model tested in this study.
Please provide more details. The entire section in the methods subsection is irrelevant.
We have made several amendments in the section of methods and write as follows:
3. METHODS
The pandemic and epidemic research context of this study was Sweden. The Swedish approach to handle SARS-CoV-2 stood out particularly in the first wave1 and has been conspicuous internationally 2,3. It was therefore a relevant point of reference and benchmark to assess in relation to other countries.
The implementation of the Swedish strategy was less strict and intrusive to the public than in many other countries4, such as Danmark, Finland and Norway, but also in comparison to most European countries. The mortality rate in Sweden per capita was among the top in the world5,6, despite the fact that it is a highly developed economy with a small population and low population density7.
An important observation is that a large part of the population is socially and culturally characterized by the norm of social distancing. The social and cultural context of Denmark, Finland and Norway resembles Sweden to a large extent8,9,10. Economic welfare is also high across these countries7. However, the governments and agencies of public health in the neighboring countries reacted and acted differently to Sweden with stricter and more intrusive measures to handle the infection rate of the SARS-CoV-2 in the society in order to protect their populations.
A closer look at the healthcare statistics between Sweden and the neighboring countries related to the SARS-CoV-2 demonstrated substantial differences5,6: (i) the number of confirmed cases was three times higher per capita in Sweden than in Denmark, Finland and Norway all together; (ii) the number of performed tests per capita in the neighboring countries together was seven times higher than Sweden; and (iii) the mortality rate was six times higher per capita in Sweden. Consequently, the morbidity per capita (i.e. the number of hospitalized and intensive care unit patients) was therefore also higher in Sweden than the neighboring countries.
The official healthcare related COVID-19 data was gathered from the Agency of Public Health11, the National Board of Health and Welfare, and the Swedish Intensive Care Registry – SIR12,13. The data bases were continuously revisited for updates throughout the first year of the pandemic
The data bases were continuously revisited for updates throughout the first year of the pandemic. We tested the research model outlined in Figure 1, with Covariance-Based Structural Equation Modelling (CB-SEM), using SPSS AMOS 26.0 software. CB-SEM enabled to shed light on the direct, indirect and mediating effects between SARS-CoV-2 related variables.
CB-SEM is a multivariate technique that enables to verify the validity and reliability of measurement and structural properties of a research model. It is a so-called full-information technique that considers the co-variances simultaneously between variables tested in the research model. CB-SEM enables to test whether there are direct and indirect effect between variables, as well as the existence of mediating effects between the same variables.
The research model was therefore divided into three sub-models: (i) direct effects; (ii) direct and indirect effects; and (iii) mediating effects. The infection rate was categorized as the independent manifest variable as it is the one which is hypothesized to cause an effect on the outcome of the others in society. Consequently, the hospitalized patients, intensive care unit patients and deceased were categorized as dependent manifest variables.
It should be noted that the infection rate at a given time does not usually lead to hospitalization, ICU or death at the same time, but there is a time-delay. A time-delay between tested Covid-19-related variables is therefore applied in the CB-SEM analyses. Nevertheless, a hypothetical situation could refer to a confirmed case on day one, being hospitalized on day two, then admitted to ICU on day three and finally, deceased on day four. In fact, all of this could take place on day one, but alternatively, with a major time-delay.
In reality, there was often a certain time-delay after a case of SARS-CoV-2 has been tested and confirmed, and the subsequent effects on the number of hospitalized and ICU patients in the near future, as well as on the time between infection until patients were deceased. In fact, the time-delay in Sweden between day of confirmed infection and: (i) patients being hospitalized – was on average 6.2 days 12; (ii) patients being submitted to ICU – was on average 10.6 days12; and (iii) until patients deceased – was on average 12.5 days 12. The SARS-CoV-2 related variables in the research model were therefore analysed with a time-delay of one week between infection rateand the other SARS-CoV-2 related variables.
- What hypotheses were set in this study and why?
We write as follows:
There were highly significant correlations between the SARS-CoV-2 related variables, with correlation coefficients ranging from 0.835 to 0.914 with p-values at 0.000. We therefore test six hypothesized relationships between the SARS-CoV-2 related variables in the research model displayed in Figure 1: (1) infection rate related positively to hospitalized patients; (2) infection rate related positively to intensive care unit patients; (3) infection rate related positively to deceased; (4) hospitalized patients related positively to deceased; (5) intensive care unit patients related positively to deceased; and (6) hospitalized patients related positively to intensive care unit patients.
- Title and purpose of the study - in my opinion, you did not emphasize the purpose in the introduction and discussion.
We have rewritten the introduction and made amendments as follows:
Ioaniddes (2022; p1) states “There are no widely accepted, quantitative definitions for the end of a pandemic such as COVID- 19. McCoy (2023; p.1) points out that: “The end of a pandemic is as much a political act as biological reality”. Furthermore, the awareness of a future pandemic may occur has become a concern. A revisit to assess the initial estimations of morbidity and mortality is therefore justified as time and timing of countermeasures are going to be crucial to monitor and control the emergence of future pandemics. The crucial question regarding the SARS-CoV-2 pandemic is therefore whether gathered knowledge and experiences have provided any lesson for sustainable health policies in the future.
There are two streams of studies reported in literature. One stream focuses on 7- and 14-days incidences to predict SARS-CoV-2 (e.g., Byambasuren et al., 2020; Sanjuán,2021; Scobie et al., 2021), while the other focuses on infection rates (e.g., Behnood et al., 2020; Manski and Molinari, 2021; Salgotra et al., 2020). To the best of the authors knowledge there are no studies that focus on comparing the predictive ability between 7- and 14-days incidences and infection rates on SARS-CoV-2 related morbidity and mortality in a given population.
The objective of this is study focuses on the lesson that can be learnt for sustainable health policies from the use of 7- and 14-days incidences versus infection rates to predict SARS-CoV-2 related morbidity and mortality in a given population. This study therefore assesses the predictive abilities of infection rates versus 7- and 14-days incidences on SARS-CoV-2 related morbidity and mortality. It also assesses the structural properties of a set of SARS-CoV-2 related variables. To the authors knowledge, there is also no previous study that has assessed the direct, indirect and mediating effects between infection rates on the one side, and morbidity and mortality on the other. The structural properties have not been prioritized, but taken for granted in terms of that infection rates influence morbidity and mortality, not the direct, indirect and mediating effects.
- Abstract and introduction - there is often repetition in the abstract and introduction section. This should be corrected, and the introduction should provide more information about the background of the study and the literature review in this area. Definitions of terms used, such as " generic research model" should be added.
We think that the abstract should summarize the paper, and therefore repetition is needed.
We have omitted the term ‘generic’.
- Figures - bad quality.
We do not understand the comment provided.
- From a researcher's perspective, I have to ask this: Why do you want to publish an article in 2024 based on results from 2021?
A lesson learnt for the future takes time to digest, and we have searched for other studies that have come up with the same or similar findings. We have not found any other study and therefore believe that the findings reported are relevant as a lesson for the future.
Please note that we have made amendments in the introduction that addresses this comment.
- I did not notice "a lesson for the future" in your manuscript (discussion).
We have rewritten and made amendments as requested.

Round 2
Reviewer 1 Report
Comments and Suggestions for Authors
Thank you for addressing my comments. Some of the newly added sentences are difficult to read and require further editing for clarity, particularly at the beginning of the introduction. Also, I suggest avoiding multiple claims that "...to the author's knowledge, there are no...etc", if done, please do it within a single paragraph.
There are several typos in the newly added content, such as Danmark, data bases, please ensure the manuscript is properly edited for English language.
I suggest further expanding the limitations section and placing it before the conclusion section following standard practices.
Comments on the Quality of English LanguageNo comments
Author Response
REVISION REPORT: Sustainability R2
Title: “Lessons for Sustainable Health Policy and the Past as Lessons for the Future”
Dear Editor, Associate Editor and Reviewers,
We are pleased to have the opportunity to make the second revision of our paper for further consideration inSustainability. We provide a detailed response (IN YELLOW) to each reviewer’s comments (IN GREEN).
We hope that you will find our revised paper much improved and satisfactory. Please note that we highlight major changes and amendments in the revised paper IN BLUE.
We believe that we have been able to address and justify our responses to all review comments appropriately.
Best regards,
The authors
Editor:
Thank you again for your manuscript submission. We noticed that this paper is still highly repeated with the following paper: https://pubmed.ncbi.nlm.nih.gov/34267295/?. Please try to lower the repetition rate during the revision stage.
We have done as advised and the similarity is significantly reduced.
Reviewer 1:
Thank you for addressing my comments. Some of the newly added sentences are difficult to read and require further editing for clarity, particularly at the beginning of the introduction.
Also, I suggest avoiding multiple claims that "...to the author's knowledge, there are no...etc", if done, please do it within a single paragraph.
It has been omitted.
There are several typos in the newly added content, such as Danmark, data bases, please ensure the manuscript is properly edited for English language.
We think most of them are the automatic formatting that make changes to the text of the paper as we could not find them in our original file. In any case, we have revised as advised.
I suggest further expanding the limitations section and placing it before the conclusion section following standard practices.
We have done as advised and write as follows:
- LIMITATIONS
This study has several limitations, such as that the COVID data gathered is limited to just one country (i.e., Sweden). The time frame of the data is also limited to the period before vaccines were introduced, although this was done to keep data unbiased. Another limitation is the absence of other COVID-related data that were not available at the time, such as virus levels in sewage. In addition, an assessment of sociographic and psychographic variables was not possible, given that data was not available that could have yielded additional insights from population characteristics. This study is also limited to one pandemic, so that comparisons are not possible. The limitations also relate to the type of virus and the effect that it had on the morbidity and mortality of populations. Although this study does have these limitations, it nonetheless offers a lesson from the past for the future, and also provides suggestions for further research.
Reviewer 2:
Dear authors,
Thank you for providing changes in the text
The comment has been noted with thanks.
Comments on the Quality of English Language : Better than in the previous version
The comment has been noted with thanks.
Reviewer 2 Report
Comments and Suggestions for Authors
Dear authors,
Thank you for providing changes in the text
Comments on the Quality of English LanguageBetter than in the previous version
Author Response

(The authors gave the same response as above.)
